# Investigation of the Behavior and Mechanism of Action of Ether-Based Polycarboxylate Superplasticizers Adsorption on Large Bibulous Stone Powder

**DOI:** 10.3390/ma14112736

**Published:** 2021-05-22

**Authors:** Zuiliang Deng, Guimin Lu, Lefeng Fu, Weishan Wang, Baicun Zheng

**Affiliations:** 1School of Resources and Environmental Engineering, East China University of Science and Technology, Shanghai 200237, China; dengzl@sunrisechem.com.cn; 2Technical Center, Shanghai Sunrise Polymer Material Co., Ltd., Shanghai 200231, China; lefeng@sunrisechem.com.cn (L.F.); weishan@sunrisechem.com.cn (W.W.); baicun@sunrisechem.com.cn (B.Z.); 3Shanghai Engineering Research Center of Construction Admixtures, Shanghai 200231, China

**Keywords:** large bibulous stone powder, polycarboxylate (PCE), adsorption, pore size

## Abstract

The aim of this paper is to study the adsorption behavior of polycarboxylate superplasticizers (PCE) on coarse aggregates with a property of high water consumption (above 2%). The coarse aggregates were ground into a powder to create large bibulous stone powder, and it was observed that significant amounts of the ether-based PCE were absorbed onto large bibulous stone powder. The adsorption rate immediately reached a maximum after 5 min and then gradually decreased until an equilibrium absorption was established after 30 min. Zeta potential, infrared spectroscopy, and thermogravimetric analysis (TGA) measurements confirmed that the polycarboxylate superplasticizer adsorbed on the surface of the stone powder. Hydrodynamic diameter measurements showed that the polycarboxylate superplasticizer molecules were smaller than pore size, and the surface area and pore volume were reduced by the polymer incorporation in the pores.

## 1. Introduction

The travel time between Maputo to Kosi Bay in Africa was drastically reduced from 6 h to 90 min when the Maputo Katembe Bridge project was finished. However, during construction, it was found that the physical and chemical properties of the local raw materials were incompatible with the ordinary admixtures used in other areas. In fact, the water absorption of coarse aggregate of the coarse stone aggregates from Mozambique was (2–5%) higher than other standard construction materials (<2%) [1].

It was surprising that the high water consumption was due to the presence of these coarse aggregates in Mozambique instead of clays from the sand, such as bentonite and kaolinite [2]. These clays are considered harmful contaminants since they significantly reduce the effectiveness of admixtures, such as water-reducing agents. One of the most popular water-reducer agents used in construction materials is polycarboxylate (PCE) which has several advantages including better dispersion, greater slump retention ability, and lower required dosages than other lignosulfonate-based, melamine-based and naphthalene-based superplasticizers, even if it possesses corrosion-inhibiting ability [3,4,5]. This overall better dispersion from PCE can be attributed to the effects of the steric hindrance from its side chains [6]. However, PCEs are adsorbed more strongly by clays than polycondensates since PCE can interact with clays by not only physisorption (surface adsorption) vis hydrogen bonding from polymers’ side chains or electrostatic adsorption from the anchoring group of polymers but also by chemisorption (intercalation) while polycondensates are only consumed by surface interactions [7,8,9,10]. In all of different types of clay, montmorillonite shows the most deleterious influence on the adsorption of PCE onto cement particles as compared with kaolinite, feldspar and mica. As such, Lei et al., Chen et al., Liu et al., and Li et al., have designed and synthesized several novel polymer structures to prevent physisorption and chemisorption onto montmorillonite, such as modifying the side chain structure, replacing it with short side chains, designing star-shaped PCE and using a zwitterionic functional group [11,12,13,14,15]. Moreover, a general trend was observed from Werani et al. that a copolymer of poly(methacrylic acid) (PMAA) and methoxy polyethylene glycols (MPEG) macromonomer with a shorter n_EO_ of 10 could provide better anti-clay dispersing ability in the cement paste contained 1% (by weight of cement) of montmorillonite [16].

Interestingly several research works show promising results on using limestone powder as a supplementary cementitious materials replacement. For instance, Felekoglu reported that a substitution of 10% of cement with quarry limestone powder (QLP) could not only improve the 28 days compressive strength of self-compacting concrete (SCCs) but also slightly reduce the required dosage of superplasticizer [17]. Moreover, Ahmadi et al. also showed that the concrete produced from partial replacement of natural zeolite could present better compressive strength but suffered from a higher dosage of superplasticizer [18]. In addition, Shi et al. also studied the mechanism of limestone powder blended with cement by researching the nucleation effect, filler effect, dilution effect and chemical effect of limestone powder. Furthermore, Li et al. presented a study that lithological stone powders (limestone and tuff) could shorten the setting time of cement paste [19]. However, insufficient studies have investigated the adsorption of PCE on to stone powders beside Lei Feng et al. and Hui Feng et al. In the study of Lei Feng et al., they measured the adsorption capacity of PCE on the surface of tuff powder with quartz, albite and potash feldspar with two methods, ultraviolet (UV) and total organic carbon (TOC) [20]. Similarly, Hui Feng et al. analyzed the impact of PCEs’ molecular structure in the presence of Ledong stone powder and Haikou stone powder. Their studies also showed that aggregates with a stratified structure and containing mica significantly increased their intercalation with PCEs and thus decreased the workability of the resulting pastes [21]. However, PCEs possessing shorter side chains could prevent such intercalation. Limited study could provide a reasonable explanation for such high water demand from large bibulous stone powder (LBSP) as in coarse aggregates used in the Maputo Katembe Bridge project in Mozambique since this specific coarse aggregate does not contain stratified structures. Here, this paper studied the PCE interactions with the coarse aggregates. First, the coarse aggregate was ground into stone powder to obtain LBSP. Then, the adsorption behavior of the ether-based PCE onto the stone powder was analyzed by the TOC method and by measuring the zeta potential in different pH and PCE concentration solutions. Finally, the adsorption mechanism of action between the stone powder and the PCE was determined by pore size distribution, Fourier-transform infrared spectroscopy (FTIR), and thermogravimetric analysis (TGA) measurements. Overall, this paper provides insights into the interactions between PCEs and coarse aggregates with high water contents.

## 2. Materials and Methods

### 2.1. Materials

A coarse aggregate with a relatively high water consumption of 3.1% was provided from the Hipermaquinas quarry in Boane District, Mozambique. The coarse aggregate was ground and sifted through a 200 mesh sieve to obtain the LBSP, as displayed in Figure 1. Then, the stone powder was dried in an oven at 105 °C for two hours, and the chemical composition was characterized using X-ray fluorescence analysis (D/MAX 2550 VB/PC diffractometer of Rigaku, Tokyo, Japan) and phase identification of such sample was measured via X-ray powder diffraction (18KW/D/max2550VB/PC, manufactured by Rigaku, Japan). As shown in Table 1 and Figure 2, the stone powder contains quartz sand, sanidine (K, Na)(Si_3_Al)O_8_), and orthoclase (KAlSi_3_O_8_). The orthoclase was mainly from magmatite, gneiss, and migmatite and could become kaolinite after weathering. 

Polycarboxylate, PCE-1, was provided by Shanghai Sunrise Polymer Material Co., Ltd., Shanghai, China. Its chemical composition is presented in Figure 3, and its characterization and solid content is shown in Table 2. 

Deionized (DI) water was obtained from a laboratory scale DI-water instrument produced by Shanghai Miaokang CD3400, Shanghai, China. The conductivity of the DI water was 0.30 μs/cm.

### 2.2. Polycarboxylate (PCE) Adsorption via Total Organic Carbon (TOC)

To ascertain the interactions between the PCE-1 and stone powder, the absorbed amounts of PCE were determined via the TOC method using a TG-WS instrument manufactured by Shanghai Lu Xiangyi Centrifuge Instrument, Shanghai, China. The standard curve for the total organic carbon with different concentrations of PCE is shown in Figure 4.

In a typical experiment, 5 g of stone powder was added to four separated Erlenmeyer flasks that contained a stir bar and 50 mL of PCE solution with different concentrations ranging from 0.1 g/L, 0.5 g/L, 1.0 g/L, and 2.5 g/L. Then, the Erlenmeyer flasks were sealed, placed in a water bath, and stirred. Next, 15 mL of the mixture was collected and centrifuged for 10 min at 10,000 rpm. Moreover, a blank sample, replacing PCE solution with a DI water, was also prepared in order to subtract the free carbon in the background. Afterwards, the supernatant was extracted and the TOC was measured on a TOC-L CPN Instrument manufactured by Shimadzu, Kyoto, Japan. 

The PCE adsorption was calculated according to: (1)$$q_{e}=\frac{\left(C_{0}-C_{e}\right)V}{m}$$
where *q_e_* (unit = mg/g) is the amount of adsorbed material; *C*_0_ and *C_e_* (unit = mg/L) are the concentration of PCE solution before and after the stone powder was added, respectively; *V* (unit = L) is the volume of sample collected for the centrifuge; and *m* (unit = g) is the mass of stone powder.

To study the adsorption kinetics, the TOC values were measured at different time points (5 min, 15 min, 30 min, 60 min, and 120 min) while for the isothermal adsorption measurements, the TOC values were measured at different temperatures (293 K, 303 K, 313 K).

### 2.3. Characterization

Supernatant from Section 2.2 with a concentration of 20 g/L was extracted and filtered by suction filtration with a filter paper. During this step, 500 mL of DI water was washed through the filter paper to remove the polymer on the surface of stone powder. Then, the filter paper was dried at 80 °C in an oven for 24 h. The collected compound, named the PCE/stone compound, was stored in desiccators for further characterizations, such as pore size distribution, FTIR, and TGA. 

#### 2.3.1. Pore Size Distribution

The pore size distribution and the Brunauer-Emmett-Teller (BET) surface area of the stone powder and compound were measured using a High-Speed Automated Surface Area and Pore Size Analyzer, ASAP2010N, manufactured by Micromeritics, Norcross, GA, USA. 

#### 2.3.2. Fourier-Transform Infrared Spectroscopy (FTIR)

In order to determine the C–H group, the PCE, LBSP, and PCE/stone compound were analyzed by FTIR (6700, Nicolet, Madison, WI, USA) over a wavelength range 400 to 4000 cm^−1^ with a resolution of 2 cm^−1^ and averaged over 32 scans. 

#### 2.3.3. Thermogravimetric Analysis (TGA)

To determine the derivative thermogravimetry (DTG) and the weight loss of the PCE/stone compound, approximately 8~12 mg of compound was placed under an N_2_ environment in a TGA device “STD Q600” from TA instrument. Test conditions: nitrogen atmosphere (100 mL/min), temperature range was 25–800 °C, and heating rate was 20 °C/min

### 2.4. Zeta Potential Measurements

The zeta potential of the suspension was measured on a Zetasizer “ZEN3600” manufactured by Malvern, Malvern, UK. The LBSP suspensions were prepared by dispersing 5.0 g of LBSP in 50 mL of PCE solutions from 0.1 g/L, 0.5 g/L, 1.0 g/L, and 2.5 g/L with the original pH value of 6.7. The suspensions were ultrasonic dispersed after 30 min, then the zeta potential measurements were conducted. The pH value of the dispersion was adjusted carefully between 2.1 and 11.0 by adding 0.1 mol/L HCl or NaOH solutions to test zeta potential with PCE concentration of 2.0 g/L. 

### 2.5. Hydrodynamic Diameter Distribution 

The hydrodynamic diameter distribution of PCE-1 in solution with concentration of 2.0 g/L was measured using a Zeta 90 Plus instrument [22]. A simulated cement pore solution (SCPS) was prepared using 1.72 g/L CaSO_4_·2H_2_O, 6.96 g/L Na_2_SO_4_, 4.76 g/L K_2_SO_4_ and 7.12 g/L KOH, the pH value of SCPS was 12.4 [23]. The measurements of PCE-1 were made in two types of solvent: DI water and SCPS. 

## 3. Results and Discussion

### 3.1. Adsorption Kinetics of PCE-1 on Stone Powder

The adsorption kinetics of four different PCE-1 concentrations (0.1 g/L, 0.5 g/L, 1.0 g/L, and 2.5 g/L) onto LBSP at a temperature of 303 K are shown in Figure 5. 

The highest adsorbed amounts of PCE were observed at 5 min, then they gradually decreased and equilibrated after 30 min. This interesting adsorption behavior can be attributed to the chemical composition of LBSP and described by diffusion theory. First, LBSP is an acidic rock and is hydrophilic because it contains 67.2% of SiO_2_, a relatively high content. Consequently, PCE, which is a strong hydrophile surfactant, easily wetted the LBSP surface. Secondly, the stone powder has a porous structure which provides many absorption sites for the PCE polymers. However, once the diffusion equilibrium (equilibrium state) was reached, some PCE polymers permeated back into the bulk solution. Hence, the adsorbed polymer amount and rate remained stable after the 30 min mark. Based on this finding, the reaction time for all subsequent experiments was fixed to 30 min. A similar measurement was taken by Hui Feng et al. There, with the PCE solution of 2.5 g/L, the adsorbed PCE amount was 4.24 mg/g and 0.79 mg/g for Ledong stone powder (LDSP) and Ledong stone powder (LDSP), respectively [21]. Similarly, Feng Lei et al. tested the adsorption of polycarboxylate superplasticizer on the surface of tuff powder with quartz, albite and potash feldspar as main components by the TOC method, and the adsorption capacity was 4.7 to 5.0 mg [20]. Here, the adsorbed amount of PCE ono stone powder (LBSP) with a PCE concentration of 2.5 g/L at 5 min mark was nearly 0.2 mg/g. Based on these results, it can be concluded that the adsorbed amount of PCE can be varied depending on the types of stone powder. 

Nevertheless, this adsorption trend from this study can be explained by diffusion theory. The steeper the concentration gradient, the faster rate of diffusion, and thus the higher the adsorption amount and the saturation capacity for PCE-1 concentrations of 0.1 g/L and 2.5 g/L are 0.051 mg/g to 0.114 mg/g, respectively. Similar adsorption kinetics were also seen in studies by Wang et al., Bulut et al., and Karagozoglu et al. where they studied absorption on clays (sepiolite or bentonite) instead of stone powder [24,25,26]. 

### 3.2. Isothermal Adsorption of PCE onto Large Bibulous Stone Powder (LBSP) 

In addition to the concentration gradient, temperature also plays a significant role in the adsorption kinetics and thermodynamics. As presented in Figure 6, the amount of sorbed PCE increased as the temperature rose, which indicated that lower temperatures did not promote polymer adsorption. Furthermore, the movement of molecules per unit time is enhanced at higher temperature, according to the so-called collision model. Thus, polymers were sorbed more effectively onto the free adsorption sites of the porous structure at higher temperatures. 

The following equations present the calculation of the change of Gibbs free energy Δ*G* (kJ/mol), enthalpy Δ*H* (kJ/mol), and entropy Δ*S* (kJ/(K·mol)):(2)$$K_{C}=\frac{C_{ads}}{C_{sol}}$$
(3)$$\Delta G=-RT\ln K_{c}$$
(4)$$\ln K_{c}=\frac{\Delta S}{R}-\frac{\Delta H}{RT}$$
where *K_C_* is the thermodynamic equilibrium constant, where *C_ads_* is adsorbed on LBSP (mol/L), *C_sol_* is the concentration in equilibrium solution (mol/L). The *ΔH* and *ΔS* can be determined by plotting ln*K_c_* against 1/*T* from the slope and intercept, respectively. All of the thermodynamic parameters are summarized in Table 3. T is the absolute temperature (*K*) and *R* is the gas constant (8.314 J/(K·mol)).

When temperature increases from 293 K to 313 K, Δ*G* slightly decreases from 13.77 kJ/mol to 13.38 kJ/mol, demonstrating that the energy barrier decreases with the increase of temperature and the adsorption reaction is more likely to occur. Chemical adsorption and physical adsorption can be distinguished by enthalpy change Δ*H* value, which is in the range of −20~400 kJ/mol for physical adsorption, and this enthalpy change is 19.47 kJ/mol for physical adsorption. The adsorption enthalpy change Δ*H* is positive, which means that the adsorption process is an endothermic process [27]. The entropy change Δ*S* is 19.52 J/(K·mol), the values of enthalpy change and entropy change are greater than 0, indicating that the adsorption reaction can be spontaneous at high temperature.

### 3.3. The Influence of pH and PCE Concentration on the Zeta Potential of LBSP

Zeta potential is a powerful tool for investigating the adsorption behavior in a biphasic system. For example, Plank et al. studied the PCE adsorption coverage of cement particles via zeta potential and concluded changes in zeta potential were an indication of high PCE adsorption [28]. Moreover, Zingg et al. suggested that the switch from positive to negative charge in cement and gypsum measured with zeta potential was evidence of PCE adsorption [29]. 

To assess the effects of these two essential factors on the interactions between PCE and LBSP, the change in the LBSP surface charge was measured at different pH and PCE concentrations. As shown in Figure 7a, the zeta potential increased from negative to positive and then decreased to an even lower surface charge as the pH rose from 2 to 11. Evidently, either strong acidic (pH = 2) or strong basic conditions (pH > 10) improved the stability of the suspension, particularly in the case of high pH values (pH = 11). 

In contrast to Figure 7b, the surface charge declined gradually with increasing PCE concentration. The increment in the zeta potential looks similar to the changes in isothermal adsorption rate in Section 3.2. Moreover, the adsorption saturation point was reached at a PCE concentration of 2.0–2.5 g/L where the change in zeta potential plateaued. It is known that PCE binding onto cement is due to chelation between the anchoring groups in the polymer and the free Ca^2+^ in solution in the cement pores [30]. As a result, only a finite amount of PCE was absorbed onto the cement since there were a finite number of Ca^2+^ cations in solution. Similarly, there are restricted binding sites since the limited free Ca^2+^ ions in the pores of the stone powder solution and, as such, the zeta potential plateaus when all of the binding sites are occupied. 

### 3.4. Characterization of the PCE/LBSP Compound

As shown in Figure 8, the FTIR spectrum of the PCE/LBSP compound was more similar to the pure LBSP as compared to pure PCE-1. However, the measured spectrum for the compound had an additional peak at −2867 cm^−1^ that belonged to the C–H bonds in the PCE-1 methylenes and supported the assertion that PCE was adsorbed on to the LBSP surface. 

If the PCE polymers were present on the surface or in the pores of the LBSP, then analysis of the TGA curves should show additional weight loss in the compounds due to the decomposition of the polymers. This method was also used in the study of Ait-Akbour et al. and Tan et al., to study intercalation of PCE on montmorillonite [31,32]. As expected in the present results, there was an additional weight loss plateau at 553~580 °C with a weight loss ratio of 0.132% in addition to the first weight loss plateau at 175~258 °C due to the water associated with the LBSP and a second weight loss plateau at 260~290 °C due to the decomposition of the LBSP crystal lattice as seen in Figure 9a,b. The extra weight loss plateau further supported the assertion that the PCE-1 had adsorbed into the LBSP pores as any polymer only bound to the surface should have been washed away during the PCE/LBSP compound preparation. 

In order to further show that PCE-1 had absorbed into the LBSP porous structure, the pore size distribution of the LBSP and PCE/ LBSP compound was analyzed. As seen in Table 4 and Figure 10, the surface area of the stone powder decreased from 21.7 nm to 6.6 nm after it was mixed with the PCE-1 solution. This pore size reduction is direct proof of the PCE-1 adsorption. 

As shown in Figure 11, the hydraulic diameter distribution of PCE-1 in an aqueous solution (DI water) varied from 4.1 nm to 8.7 nm, with the maximum diameter of 5.6 nm. In contrast, in SCPS, the hydrodynamic diameter distribution ranged from 4.1 nm to 11.7 nm, where the maximum diameter was 7.5 nm. This result shows that the PCE-1 polymer could enter the voids in the LBSP particles since the polymer had a smaller hydrodynamic diameter than the LBSP pore size. As a consequence, the BJH pore volume of PCE/LBSP compound was reduced.

## 4. Conclusions

The interactions between large bibulous stone powder and ether-based PCE were studied. Significant amounts of PCE were sorbed both onto the surface and into the pores of the LBSP in the first 5 min, then the adsorption rate gradually decreased. Finally, when PCE polymers fully covered the surface and pores, an adsorption equilibrium was reached after 30 min. The adsorption process of PCE onto LBSP was an endothermic process, thus the equilibrium adsorption capacity increased with the increase of temperature. The hypothesized adsorption mechanism was first studied with zeta potential measurements where the zeta charge became more negative after an addition of the PCE solution. Also, the FTIR spectrum for the PCE/LBSP compound displayed the C–H functional groups from the methylenes in the ether-based PCE polymer. Furthermore, the PCE/LBSP compound showed a reduction of both the BET surface area and pore size due to the presence of PCE in the LBSP pores, and the hydrodynamic diameter of PCE-1 was smaller than the pore size of LBSP. Moreover, TGA experiments also showed evidence of an additional weight loss plateau at 553~580 °C in the compound due to the degradation of the absorbed polymers.

The results of this study provide an important theoretical basis for the interaction between PCE and porous aggregate, based on which, different molecular structure PCE will be designed to improve the application performance in concrete with porous aggregate.

## Figures and Tables

**Figure 1 materials-14-02736-f001:**
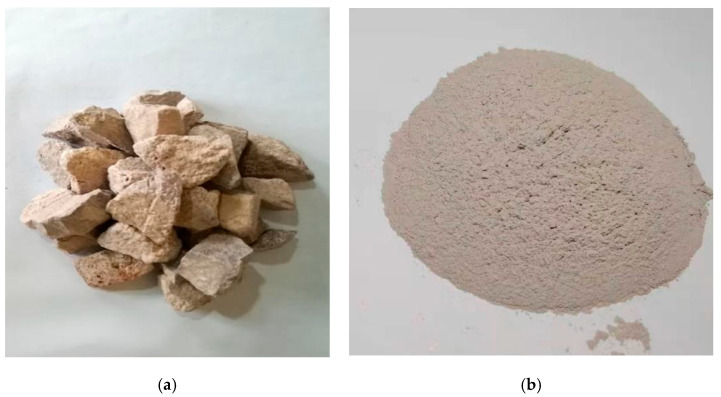
Optical photographs of coarse aggregate (**a**) and large bibulous stone powder (LBSP) (**b**).

**Figure 2 materials-14-02736-f002:**
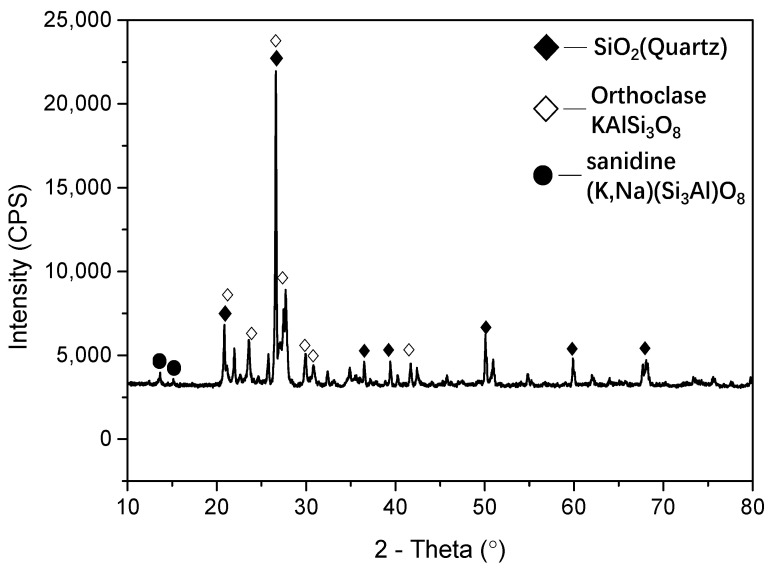
X-ray diffraction (XRD) patterns of stone powder.

**Figure 3 materials-14-02736-f003:**
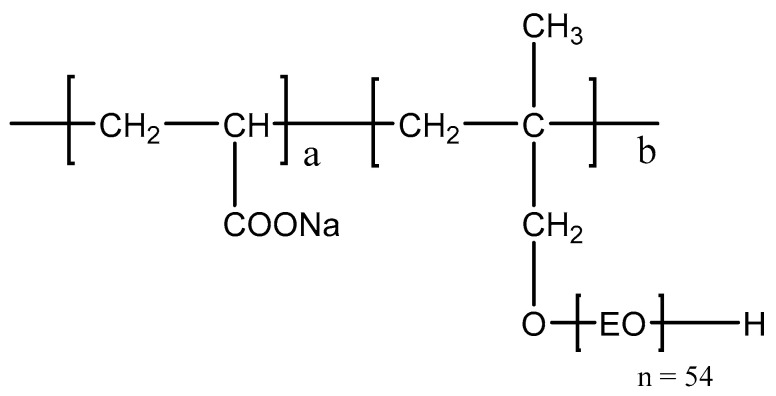
Molecular structure of the ether-based polycarboxylate (PCE) used in this study.

**Figure 4 materials-14-02736-f004:**
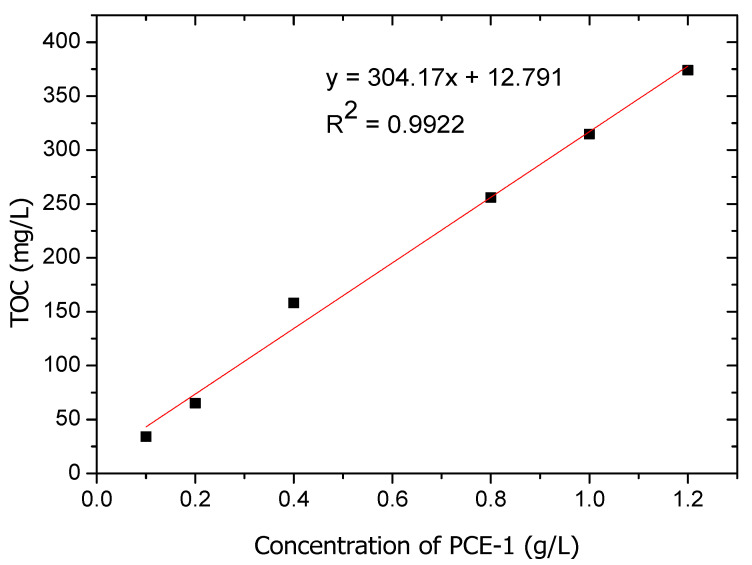
The adsorption standard curve of PCE-1.

**Figure 5 materials-14-02736-f005:**
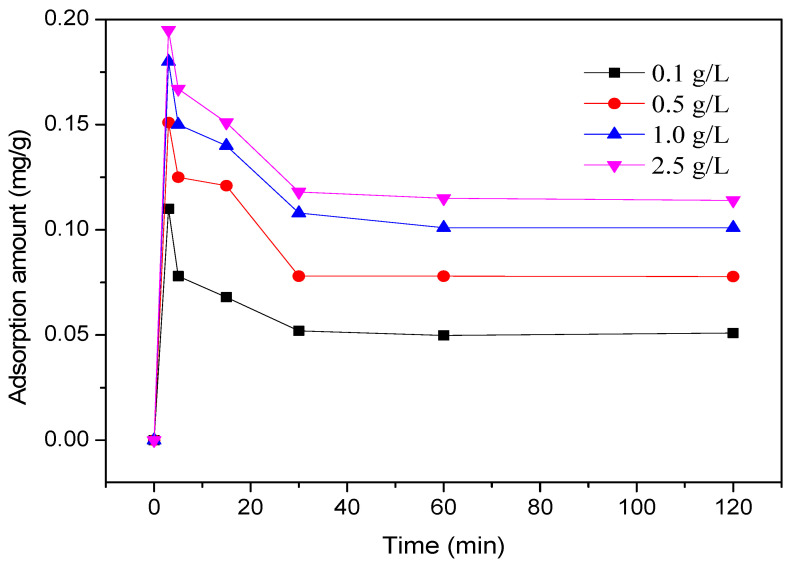
Adsorption behavior of four different PCE-1 concentrations in the stone powder over 120 min.

**Figure 6 materials-14-02736-f006:**
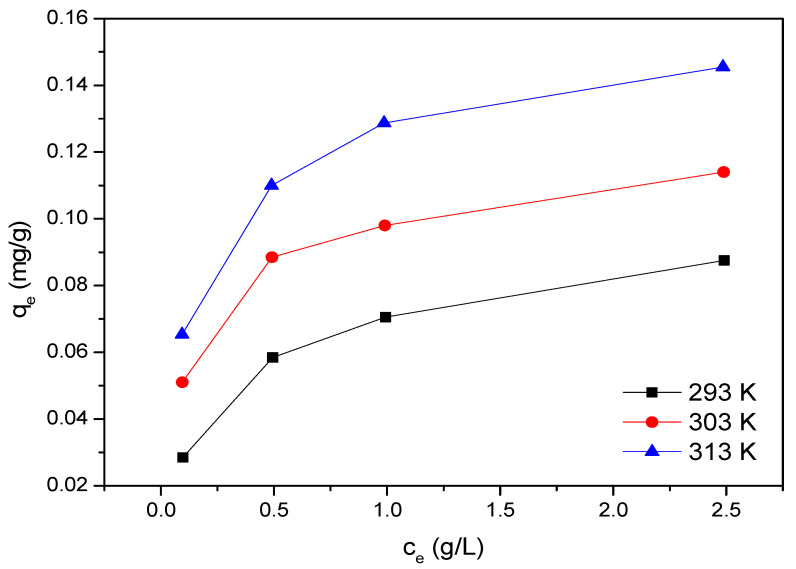
Adsorption isotherms of PCE-1 on LBSP at three different temperatures. (293 K, 303 K, and 313 K) as indicated in the legend.

**Figure 7 materials-14-02736-f007:**
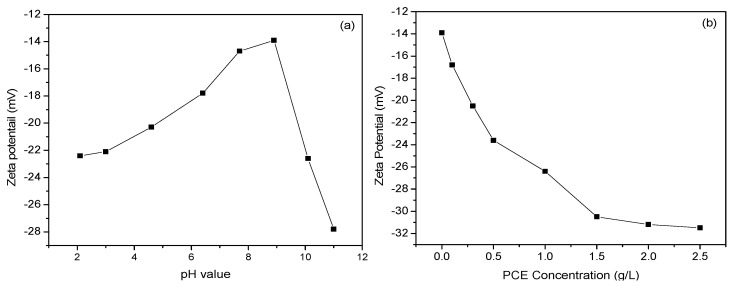
Effect of pH and PCE concentration on the measured zeta potentials of LBSP. (**a**) pH values. (**b**) PCE concentration.

**Figure 8 materials-14-02736-f008:**
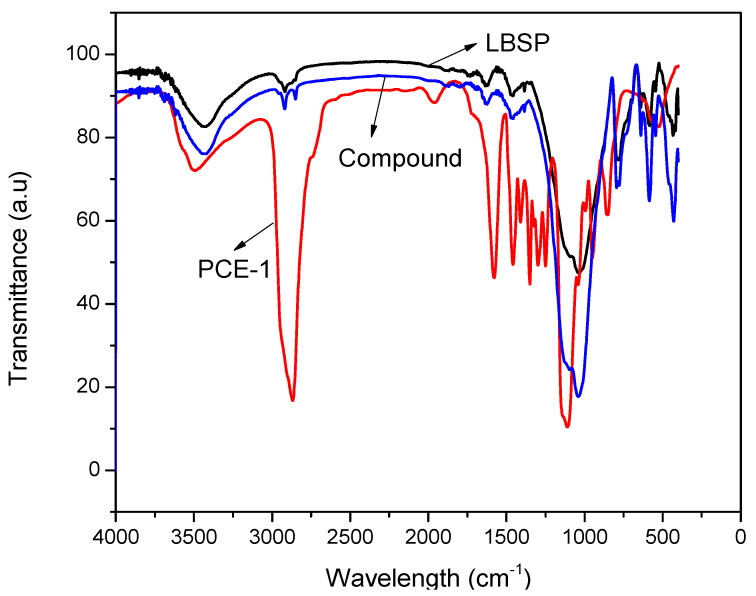
Fourier-transform infrared (FTIR) spectra of PCE-1, LBSP, and the PCE/LBSP compound.

**Figure 9 materials-14-02736-f009:**
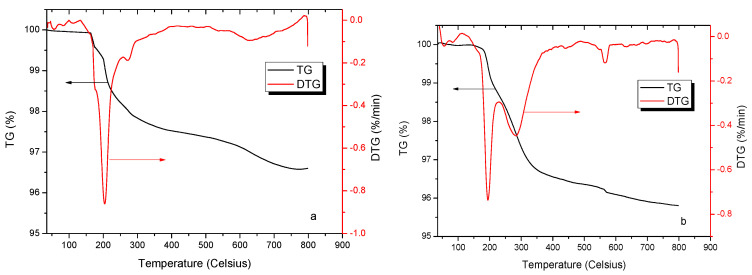
Thermogravimetry/derivative thermogravimetry (TG/DTG) curves of the LBSP and compound (**a**) LBSP; (**b**) PCE/LBSP compound.

**Figure 10 materials-14-02736-f010:**
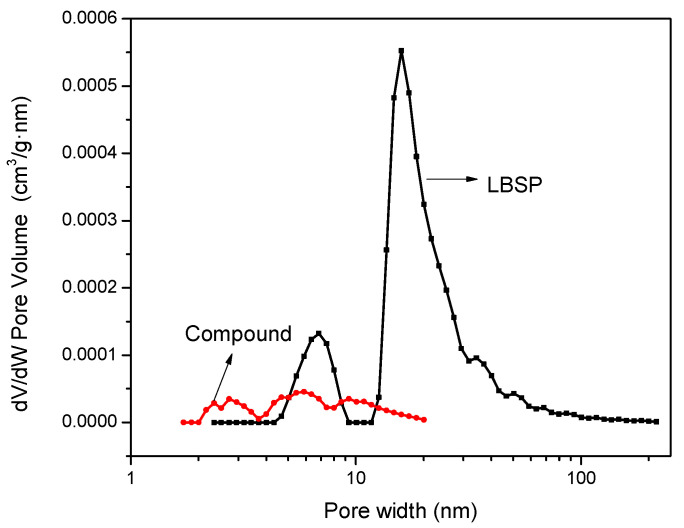
The derivative in pore volume, dv/dw, in the LBSP and PCE/LBSP compound.

**Figure 11 materials-14-02736-f011:**
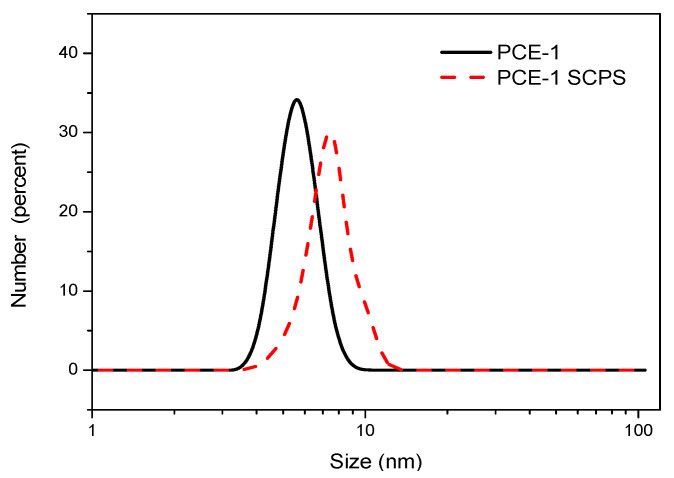
Hydrodynamic diameter of PCE-1 in water and in the simulated stone powder solution (SCPS).

**Table 1 materials-14-02736-t001:** Chemical composition of large bibulous stone powder.

Chemical Composition (wt%)												
SiO_2_	Al_2_O_3_	Fe_2_O_3_	Na_2_O	CaO	TiO_2_	BaO	ZrO_2_	SO_3_	Br	MgO	MnO	P_2_O_5_
67.22	13.25	5.17	2.18	1.20	0.33	0.14	0.13	0.12	0.08	0.06	0.06	0.04

**Table 2 materials-14-02736-t002:** Characteristic molecular parameters of PCE-1.

Sample	Length of Side Chain (n_EO_)	Side Chain Density (a:b)	M_n_ (g/mol)	M_w_ (g/mol)	Polydispersity Index (M_w_/M_n_)	Solid Content (wt%)
PCE-1	54	3.2:1	18,600	34,800	1.87	50.0

**Table 3 materials-14-02736-t003:** Thermodynamic parameters of polycarboxylate water reducer onto LBSP.

Temperature/K	*K_c_*(×10^–3^)	Δ*G*/(kJ/mol)	Δ*H*/(J/mol)	Δ*S*/(J/(K·mol))
293	0.351	13.77	19.47	19.52
303	0.458	13.57		
313	0.585	13.38		

**Table 4 materials-14-02736-t004:** Pore size distribution in the LBSP and PCE/ LBSP compound via the Barrett–Joyner–Halenda (BJH) method.

Powder Type	BET Surface Area/m²/g	Average Pore Diameter /nm	BJH Volume of Pores in 1.7–30 nm/1*10^−3^ cm³/g	Adsorption Surface Area of Pores in 1.7–30 nm/cm^2^/g
Stone powder	4.58	21.7	8.474	0.749
Compound	1.55	6.6	0.792	0.262

## Data Availability

Data sharing is not applicable to this article.

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
