# Peer review of "Investigation of the Behavior and Mechanism of Action of Ether-Based Polycarboxylate Superplasticizers Adsorption on Large Bibulous Stone Powder"

_materials, 2021, doi:10.3390/ma14112736_

Round 1

Reviewer 1 Report

The aim of this paper is to study the adsorption behavior of polycarboxylate (PCE) copolymers on coarse aggregates with a property of high-water consumption (above 2%).

  1. When analyzing the composition of the sample (large bibulous stone powder) used methods of X-ray phase analysis and X-ray fluorescence method. In the experimental part of the work should give the names of the instruments and the country of manufacture.
  2. How was the solid-liquid ratio determined when studying the sorption of PCE-1 onto LBSP? The kinetics of the sorption process should be analyzed using different kinetic models.
  3. The study obtained adsorption isotherms of PCE-1 on LBSP at three different temperatures. (293K, 303K, 313K). The activation energy of the investigated process should be calculated.
  4. Some inaccuracies in the text should be checked and corrected. For example, line 109...Where is section 1.2 presented?
  5. The conclusions should be revised in relation to the applied relevance of the study.

Author Response

Point 1: When analyzing the composition of the sample (large bibulous stone powder) used methods of X-ray phase analysis and X-ray fluorescence method. In the experimental part of the work should give the names of the instruments and the country of manufacture. 

Response 1: We have added this information, please see the change at line 94 to line 95 from the revised paper.

Point 2: How was the solid-liquid ratio determined when studying the sorption of PCE-1 onto LBSP? The kinetics of the sorption process should be analyzed using different kinetic models.

Response 2: Thank you for your review. We have determined the solid-liquid ratio through preliminary exploratory experiments that the adsorbed amounts of PCE onto stone powder was not as much as that onto clay (8-10 mg/g of clay). Therefore, we have decided to use a higher solid-liquid ratio in order to achieve more reasonable data (lower error).

The analysis of kinetics of the sorption process using different kinetic models is very important for adsorption behaviour, Wang et al. investigated the kinetics for adsorption of PVA onto bentonite, and analysed the sorption process using three kinetic models, the adsorption process obeyed pseudo-second-order kinetic model. We will improve this part of the content in subsequent research.

Point 3: The study obtained adsorption isotherms of PCE-1 on LBSP at three different temperatures. (293K, 303K, 313K). The activation energy of the investigated process should be calculated.

Response 3: Thanks for your suggestion. The activation energy of the investigated process has been added in the article, please line 219 to line 237 from the revised paper, as shown in the below:

The following equations present the calculation of the change of Gibbs free energy △G (kJ/mol), enthalpy △H (kJ/mol), and entropy △S (kJ/( Kžmol))

(1)

(2)

(3)

where KC is the thermodynamic equilibrium constant, where Cads is adsorbed on LBSP (mol/L), Csol is the concentration in equilibrium solution (mol/L). The △H and △S can be determined by plotting lnKc against 1/T from the slope and intercept, respectively. All of the thermodynamic parameters are summarized in Table 3. T is the absolute temperature (K) and R is the gas constant (8.314 J/( Kžmol)).

When temperature increases from 293K to 313K, △G slightly decreases from 13.77kJ /mol to 13.38kJ/mol, demonstrating that the energy barrier decreases with the increase of temperature and the adsorption reaction is more likely to occur. Chemical adsorption and physical adsorption can be distinguished by enthalpy change △H value, which is in the range of - 20 ~ 400 kJ / mol for physical adsorption, and this enthalpy change is 19.47 kJ / mol for physical adsorption. The adsorption enthalpy change ∆H is positive, which means that the adsorption process is an endothermic process. The entropy change △S is 19.52 J/( Kžmol), the values of enthalpy change and entropy change are greater than 0, indicating that the adsorption reaction can be spontaneous at high temperature.

Table 3  Thermodynamic parameters of polycarboxylate water reducer onto LBSP

Temperature/K

Kc(×10-3)

â–³G/(kJ/mol)

â–³H/(J/mol)

△S/(J/(Kžmol))

293

0.351

13.77

19.47

19.52

303

0.458

13.57

313

0.585

13.38

Point 4: Some inaccuracies in the text should be checked and corrected. For example, line 109...Where is section 1.2 presented?

Response 4: We are sorry for our negligence. We have changed section 1.2 to section 2.2 , please see line 140 form from the revised paper.

Point 5: The conclusions should be revised in relation to the applied relevance of the study.

Response 5: Thanks for your suggestion. we have remodified the conclusion, please see line 306 to line308 and line 317 to line 319 as shown below:

The adsorption process of PCE onto LBSP was an endothermic process, thus the equilibrium adsorption capacity increased with the increase of temperature.

The results of this study provide an important theoretical basis between PCE and porous aggregate, based on which, different molecular structure PCE will be designed to improve the application performance in concrete with porous aggregate.

Reviewer 2 Report

The study carried out is interesting. However, some improvements should be made before the paper deserves publication:

  • Section 2.2: Why did not you use a reference solution (without PCE)?
  • Section 2.3.3: Indicate the pH value of the SCPS.
  • Section 3.2: Compare the obtained results with previous ones already published.
  • Section 3.3: It should be interesting take into account higher pH values (>12.5) that are the typical pH values of the pore solution of a cementitious material.
  • Section 3.4: What PCE-1 concentration are you considering in the results shown? Why do not you consider all the PCE-1 concentrations studied? For example, is the decrease in the porosity values higher for higher PCE-1 concentrations? This study is needed to corroborate the conclusions obtained.

Author Response

Point 1: Section 2.2: Why did not you use a reference solution (without PCE)?

Response 1: Thank you for pointing out, in fact, we should have mentioned that all TOC measurements were subtracted with TOC values obtained from a blank sample, a reference solution (the one without PCE solution but DI water). We also added a sentence to be clarified, please see line 125 to line 127 from the revised paper, as shown below:

“Moreover, a blank sample, replacing PCE solution with a DI water, was also prepared in order to subtract the free carbon in the background.”

Point 2: Section 2.3.3: Indicate the pH value of the SCPS.

Response 2: Thank you again for pointing it out. The pH value of the SCPS is 12.4, and we have indicated this value in the paper as well, please see line 172 from the revised paper.

Point 3: Section 3.2: Compare the obtained results with previous ones already published.

Response 3: Thanks for your review. We have added more literatures for the discussion, please see line 192 to line 200 from the revised paper, as shown below:

A similar measurement was done by Hui Feng et al. There, with the PCE solution of 2.5 g/L, the adsorbed PCE amount was 4.24 mg/g and 0.79 mg/g for Ledong stone powder (LDSP) and Ledong stone powder (LDSP), respectively. Similarly, Feng Lei et al. tested the adsorption of polycarboxylate superplasticizer on the surface of tuff powder with quartz, albite and potash feldspar as main components by TOC method, and the adsorption capacity was 4.7 to 5.0 mg. And, here, the adsorbed amount of PCE ono stone powder (LBSP) with a PCE concentration of 2.5 g/L at 5 mins mark is nearly 0.2 mg/g. Based on these results, it can be concluded that the adsorbed amount of PCE can be varies depending on the types of stone powder.

Point 4: Section 3.3: It should be interesting take into account higher pH values (>12.5) that are the typical pH values of the pore solution of a cementitious material.

Response 4: First of all, thank you for your comment. We will put related experiment results in the future research since we have already run out of the raw materials supply.

However, based on our understanding, it is likely that the lower pH (<12) will reduce the edge charges of LBSP (more positively charged) since it contains in rich of SiO2. Additionally, with pH higher than 12, the edge charges of LBSP will have more negative sites. The reason is because of OH- from the SiO2 will disassociate in H2O, and high pH will increase the concentration of H+ ions from this disassociation reaction.

Point 5: Section 3.4: What PCE-1 concentration are you considering in the results shown? Why do not you consider all the PCE-1 concentrations studied? For example, is the decrease in the porosity values higher for higher PCE-1 concentrations? This study is needed to corroborate the conclusions obtained.

Response 5: Thank you for your comments. In this experiment, we have selected the highest concentration of PCE-1 to test, to ensure that the adsorption is saturated, but we can carry out the experiments of different concentrations of PCE-1 in the future, and combine the changes of pore structure with the previous experiments.

Reviewer 3 Report

The manuscript is well written and scientifically sound. This work can be accepted after a minor revision. There are a couple of comments from the reviewer.

  1. This work is entirely based on the experimental approach. In Section 2, there should be some illustrations explaining the experimental procedures to make the paper more understandable. Also, it will be much better to put some photos (materials, specimens, experimental apparatus) and any photos/figures for Section 2.3 Characterization.
  2. A more in-depth literature review is needed. There is no reference published after 2018 (This work is even published in Chinese). More recent research has to be reviewed. Also, in line 43, Lei et al., Xiao et al., and Li et al. do not match with the references [11-15].

Author Response

Point 1:This work is entirely based on the experimental approach. In Section 2, there should be some illustrations explaining the experimental procedures to make the paper more understandable. Also, it will be much better to put some photos (materials, specimens, experimental apparatus) and any photos/figures for Section 2.3 Characterization.

Response 1: Thanks for your suggestion. We have added Figure 1 of material in revised paper. And for experimental procedures, we have added more details from Line 139 to line 173 as below:

2.3. Characterization

Supernatant from Section 2.2 with a concentration of 20 g/L was extracted and filtered by suction filtration with a filter paper. During this step, 500 mL of DI water was washed through the filter paper to remove the polymer on the surface of stone powder. Then, the filter paper was dried at 80 ℃ in an oven for 24 hours. The collected compound, named the PCE/stone compound, was stored in desiccators for further characterizations, such as Pore Size Distribution, FTIR, and TGA.

2.3.1. Pore Size Distribution

The pore size distribution of the stone powder and compound were measured using a High Speed Automated Surface Area and Pore Size Analyzer, ASAP2010N, manufactured by Macrometrics, USA.

2.3.2. Fourier-transform infrared spectroscopy (FTIR)

In order to determine the C-H group, the PCE, LBSP, PCE/stone compound were analyzed by FTIR (6700, Nicolet) over a wavelength range 400 to 4000 cm-1 with a resolution of 2 cm-1 and averaged over 32 scans.

2.3.3. Thermogravimetric analysis (TGA)

To determine the derivative thermogravimetry (DTG) and the weight loss of the PCE/stone compound, approximately 8~12 mg of compound was placed under an N2 environment in a TGA device “STD Q600” from TA instrument. Test conditions: nitrogen atmosphere (100 ml / min), temperature range was 25-800 ℃, heating rate  was 20 ℃/ min

2.4. Zeta potential measurements

The zeta potential of the suspension was measured on a Zetasizer “ZEN3600” manufactured by Malvern. The LBSP suspensions were prepared by dispersing 5.0 g of LBSP in 50 mL of PCE solutions from 0.1 g/L, 0.5 g/L, 1.0 g/L, and 2.5 g/L with the original pH value of 6.7. The suspensions were ultrasonic dispersed after 30min, then the zeta potential measurements were conducted. The pH value of the dispersion was adjusted carefully between 2.1 and 11.0 by adding 0.1 mol/L HCl or NaOH solutions to test zeta potential with PCE concentration of 2.0 g/L.

2.5. Hydrodynamic diameter distribution

The hydrodynamic diameter distribution of PCE-1 in solution with concentration of 2.0 g/L was measured using a Zeta 90 Plus instrument. A simulated cement pore solution (SCPS) was prepared using 1.72 g/L CaSO4·2H2O, 6.96 g/L Na2SO4, 4.76 g/L K2SO4 and 7.12 g/L KOH, the pH value of SCPS was 12.4. The measurements of PCE-1 were made in two types of solvent: DI water and SCPS.

Point 2: A more in-depth literature review is needed. There is no reference published after 2018 (This work is even published in Chinese). More recent research has to be reviewed. Also, in line 43, Lei et al., Xiao et al., and Li et al. do not match with the references

Response 2: Thank you for your comment, we have fixed the problem on line 50, and also added more literature review in revised paper from line 44 to line 77 as shown below:

In all of different types of clay, montmorillonite shows the most deleterious influence on the adsorption of PCE onto cement particles as compared with kaolinite, feldspar and mica. As such, Lei et al., Chen et al., Liu et. al and Li et. al. have designed and synthesized several novel polymer structures to prevent physisorption and chemisorption onto montmorillonite, such as modifying the side chain structure, replace with short side chains, design star-shaped PCE and using zwitterionic functional group. Moreover, a general trend was observed from Werani et al. that a copolymer of poly(methacrylic acid) (PMAA) and methoxy polyethylene glycols (MPEG) macromonomer with a shorter nEO of 10 could provide better anti-clay dispersing ability in the cement paste contained 1% (by weight of cement) of montmorillonite.

Interestingly several researches show promising results on using limestone powder as a supplementary cementitious materials replacement. For instance, Felekoglu reported a substitution of 10% of cement with quarry limestone powder (QLP) could not only improve the 28 days compressive strength of self-compacting concrete (SCCs) but also slightly reduce the required dosage of superplasticizer. Moreover, Ahmadi et al. also showed that the concrete produced from partial replacement of natural zeolite could present better compressive strength but suffer from a higher dosage of superplasticizer. In addition, Shi et al. also studied the mechanism of limestone powder blended with cement by researching nucleation effect, filler effect, dilution effect and chemical effect of limestone powder. Furthermore, Li et al. presented a study that lithological stone powders (limestone and tuff) could shorten the setting time of cement paste. However, insufficient studies have investigated the adsorption of PCE onto stone powders beside Lei Feng et al and Hui Feng et al. In the study of Lei Feng et al., they measured the adsorption capacity of PCE onto the surface of tuff powder with quartz, albite and potash feldspar with two methods, UV and TOC. Similarly, Hui Feng et al. analyzed the impact of PCEs’ molecular structure in the presence of Ledong stone powder and Haikou stone powder. Their studies also showed that aggregates with a stratified structure and containing mica significantly increased their intercalation with PCEs and thus decreased the workability of the resulting pastes. However, PCEs possessing shorter side chains could prevent such intercalation. Limited study could provide a reasonable explanation on such high-water demand from large bibulous stone powder (LBSP) as coarse aggregates used in the Maputo Katembe Bridge project in Mozambique since this specific coarse aggregate does not contain stratified structures.

Round 2

Reviewer 1 Report

Dear Authors! Thank you for your detailed responses to the comments.

Reviewer 2 Report

The manuscript has been improved.

Reviewer 3 Report

The authors have put much effort to address all the issues brought by the reviewer. The manuscript should be accepted for publication in the current form.